# Federated Recommendation with Reinforcement Learning based Knowledge Distillation

## Abstract

Federated recommendation (FR) has emerged as a promising paradigm to enhance user privacy by training models in a distributed manner, where participants share model updates instead of raw data. Despite its advantages, three major challenges remain: (i) Privacy leakage. Direct parameter sharing risks exposing model information. (ii) Degraded performance. Parameter-transfer based methods often underperform compared to centralized training due to limited knowledge exchange. (iii) Communication overhead. Frequent synchronization of large models incurs prohibitive communication costs. To address these challenges, we propose **FedKDRec**, a novel FR framework based on bidirectional knowledge distillation (KD) with agents. Instead of exchanging parameters, FedKDRec transfers soft predictions under a privacy-preserving mechanism, thereby protecting both user data and model assets. To further improve effectiveness and efficiency, we introduce: (1) a server-oriented agent that dynamically assigns weights to client knowledge in multi-teacher KD, and (2) a client-oriented agent that selectively transfers informative yet lightweight samples from the server. Extensive experiments across diverse datasets and models demonstrate that FedKDRec significantly achieves superior performance and reduces communication overhead compared to parameter-transfer FR methods and existing KD-based baselines.

## 1 Introduction

Recommender systems play a crucial role in predicting user preferences from massive user-generated data. To achieve high effectiveness, many institutes rely on aggregating private data from users to train powerful recommendation models. However, with the growing concerns over user privacy and the enforcement of strict data protection regulations such as the EU General Data Protection Regulation (Voigt & Von dem Bussche, 2017), traditional centralized training protocols have become increasingly infeasible due to the risk of privacy leakage.

The conflict between the demand for accurate personalization and the obligation to protect sensitive data necessitates new training paradigms. In this context, federated recommendation (FR) (Yang et al., 2020) emerges as a promising alternative, enabling users to participate as clients that collaboratively train a global model on a central server while keeping raw data local and inaccessible to the server, so that these parameter-transfer FR methods can achieve privacy-preserving recommendation (Zhang et al., 2023a; Agrawal et al., 2024; Wang et al., 2025; Li et al., 2024b).

Despite the recent advances of parameter-transfer FR, several critical challenges remain: (i) privacy preservation in FR extends beyond protecting raw user data, as the recommendation models themselves represent critical intellectual property-particularly the global model, which aggregates collective user preferences-yet safeguarding model privacy has received comparatively less attention (Ye et al.; Zhu et al., 2023; Wei et al., 2020; Zhao et al., 2025); (ii) although parameter-transfer based FR methods are theoretically convergent (Sahu et al., 2018; Karimireddy et al., 2020), they often underperform compared to centralized training, leading to a persistent performance gap that hinders practical adoption; and (iii) real-world recommendation models are parameter-heavy, often incorporating deep neural networks and embeddings for millions of items (Li et al., 2024a), and while parameter-efficient strategies exist, parameter-transfer methods still face a fundamental trade-off between communication efficiency and model performance (Huang et al., 2024; Zheng et al., 2024), which constrains their scalability.

To this end, we propose a knowledge distillation (KD) (Hinton et al., 2015) based FR framework that directly addresses the aforementioned challenges by transferring knowledge rather than raw data, labels, or model parameters, thereby protecting both user data and model assets. In our design, clients share soft predictions generated by their local models with the server, while the server returns predictions from the global model to the clients. Compared with parameter-transfer approaches, KD-based FR is inherently more communication-efficient, as predictions are typically lower dimensional than full model parameters. Furthermore, to overcome the limitation of transferring predictions, we introduce a mixed-label distillation strategy, where soft predictions are combined with hard labels. This hybrid design enriches the transferred knowledge with both predictive uncertainty and factual supervision, ultimately leading to more accurate and robust recommendation performance.

To be specific, we develop FedKDRec, a KD-based FR framework that incorporates bidirectional KD between clients and the server. To further address the challenges and enhance recommendation quality while reducing communication overhead, particularly for resource constrained clients, we design two specialized agents within FedKDRec. (1) **Clients-to-server KD agent.** During knowledge transfer to the server, this agent adopts reinforcement learning (RL) based multi-teacher KD (Zhuang et al., 2025; Yu et al., 2025; Hossain et al., 2025), where each client serves as a teacher. It dynamically assigns teacher weights according to their effectiveness on the server model, thereby improving global recommendation performance. (2) **Server-to-clients KD agent.** During knowledge transfer to clients, this agent leverages an actor-critic framework (Sutton et al., 1999) to adaptively select informative samples, which enriches the transferred knowledge while reducing communication costs. Importantly, both agents operate exclusively during the training phase of FR. They neither introduce additional parameters into the recommendation models nor incur extra computation during inference, ensuring the practicality and efficiency of the proposed framework. In a nutshell, the key contributions of this study can be outlined as follows:

- We incorporate a KD mechanism into FR by enabling bidirectional knowledge transfer between clients and the server. This design facilitates joint model training while preserving client data privacy and the server model.

- For clients-to-server KD, we integrate RL-based multi-teacher KD. Specifically, a server-oriented agent interacts with the multi-teacher KD environment to dynamically generate teacher weights for robust aggregation, thereby improving the performance.

- For server-to-client KD, we design a client-oriented agent that focuses on efficiency by selecting the most valuable knowledge samples, enhancing the quality of transferred knowledge while simultaneously reducing communication cost.

- Our approach is designed to be compatible with existing FR models, serving as a plug-and-play enhancement. Extensive experiments across diverse datasets and models demonstrate the approach's effectiveness in consistently enhancing both recommendation and efficiency.

## 2 PRELIMINARIES

**Federated Recommendation (FR).** In FR, let $\mathcal{V}$ and $\mathcal{U}$ denote the item and user (client) sets with sizes $|\mathcal{V}|$ and $|\mathcal{U}|$, respectively. Throughout this paper, we use the terms user and client interchangeably, since each client corresponds to a unique user. Each client $u_i \in \mathcal{U}$ retains only its private dataset $\mathcal{D}_i = \{(u_i, v_j, r_{i,j})\}_{v_j \in \mathcal{V}_i}$, where $r_{i,j} = 1$ denotes a positive interaction between user $u_i$ and item $v_j$, and $r_{i,j} = 0$ indicates a negative interaction. In each training round $t$, the central server samples a subset of clients $\mathcal{U}_t \subseteq \mathcal{U}$ according to a predefined sampling ratio $\mathcal{C}$ to collaboratively optimize the clients' models and update the server-side global model.

**Multi-teacher Knowledge Distillation (KD).** We begin with standard KD, where a teacher model $\mathbf{M}^T$ transfers knowledge to a student model $\mathbf{M}^S$. The teacher $\mathbf{M}^T$ is trained on hard (ground-truth) labels $r$ and produces soft labels (teacher predictions) $r^T$. The student $\mathbf{M}^S$ is then trained with both $r$ and $\hat{r}^T$ under the following loss:

$$L_{KD} = \alpha L_{ce}(\tilde{r}^S, r) + (1 - \alpha)D(\tilde{r}^S, \hat{r}^T), \qquad (1)$$

where $L_{ce}$ is the cross-entropy loss on hard labels, $D$ measures the discrepancy between $\tilde{r}^S$ and $\hat{r}^T$ (e.g., KL divergence or cross-entropy), and $\alpha$ controls the weights of the two.

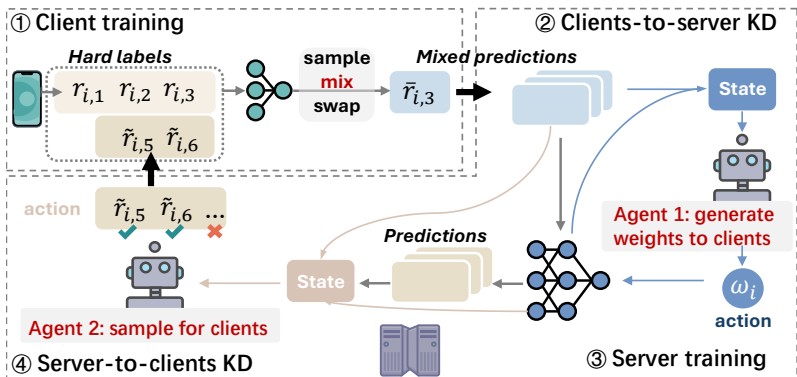

Figure 1: The framework of FedKDRec.

In multi-teacher KD, a set of teachers $\mathcal{T} = \{\mathbf{M}^{T_i}\}_{i=1}^{|\mathcal{T}|}$ transfer knowledge to $\mathbf{M}^S$. The loss extends:

$$L_{KD} = \alpha L_{ce}(\tilde{r}^S, r) + (1 - \alpha) \sum_{i=1}^{|\mathcal{T}|} \omega_i D(\tilde{r}^S, \hat{r}^{T_i}), \tag{2}$$

where $\{\omega_i\}_{i=1}^{|\mathcal{T}|}$ are teacher weights, the optimizing objective in multi-teacher KD (Yang et al., 2025).

## 3 METHODOLOGY

### 3.1 AN OVERVIEW OF FEDKDREC

Our proposed FedKDRec is a KD-based FR method which transfers soft labels rather than model parameters (Sun et al., 2025; Zhang et al., 2024). The overall framework is described in Figure 1.

**Client training.** In each round $t$, for each selected client $u_i \in \mathcal{U}_t$, the local model is trained on two datasets: 1) the private dataset $\mathcal{D}_i = \{(u_i, v_j, r_{i,j})\}_{v_j \in \mathcal{V}_i}$ with hard labels $r_{i,j}$, and 2) the auxiliary dataset $\tilde{\mathcal{D}}_i = \{(u_i, v_j, \tilde{r}_{i,j})\}_{v_j \in (\tilde{\mathcal{V}}_i \setminus \mathcal{V}_i)}$ where $\tilde{\mathcal{V}}_i$ and their soft labels are provided by the server ($\tilde{\mathcal{D}}_i = \emptyset$ when $t = 0$). Then $u_i$ updates model $\mathbf{M}_{i,t+1}$ by minimizing the loss:

$$\sum_{(u_i,v_j,r_{i,j}) \in \mathcal{D}_i \cup \tilde{\mathcal{D}}_i} L_{ce}(\hat{r}_{i,j}, r_{i,j}) = - \sum_{(u_i,v_j,r_{i,j}) \in \mathcal{D}_i \cup \tilde{\mathcal{D}}_i} r_{i,j} \log(\hat{r}_{i,j}) + (1 - r_{i,j}) \log(1 - \hat{r}_{i,j}), \tag{3}$$

where $\hat{r}_{i,j} = \mathbf{M}_{i,t}(u_i, v_j)$.

**Clients-to-server KD.** After local training, client $u_i$ generates predictions $\hat{r}_{i,j} = \mathbf{M}_{i,t+1}(u_i, v_j)$ and applies *sample*, *mix*, and *swap* strategies to transfer knowledge while protecting privacy.

First, client $u_i$ randomly samples items $\hat{\mathcal{V}}_i \leftarrow sample(\mathcal{V}_i \cup \tilde{\mathcal{V}}_i \mid \beta_{i,t})$ with sampling ratio $\beta_{i,t}$. This randomization satisfies $(\epsilon, \delta)$ differential privacy and effectively mitigates top-guess attacks, as demonstrated and evaluated in (Sun & Lyu, 2021; Yuan et al., 2024).

Next, to alleviate knowledge loss compared with the common KD setting as Eq 1, for items from the private dataset with hard labels, predictions are mixed with hard label values:

$$\overline{r}_{i,j} = \alpha r_{i,j} + (1 - \alpha)\hat{r}_{i,j}, \; v_j \in \hat{\mathcal{V}}_i \cap \mathcal{V}_i, \tag{4}$$

where $\alpha$ controls the weights. Compared with directly transmitting raw predictions, this label mixing strategy helps retain more informative knowledge (as Appendix C.1).

Besides, a swapping mechanism further enhances privacy. Specifically, the client randomly selects a portion $\lambda$ of items with high values predictions and exchange their labels with negative samples whose $r_{i,j} = 0$, forming the final uploaded dataset $\hat{\mathcal{D}}_i$ as: $\hat{\mathcal{D}}_i \leftarrow swap(\hat{\mathcal{D}}_i \mid \lambda)$.

**Server training.** The server updates $\mathbf{M}_{s,t+1}$ using the received datasets $\{\hat{\mathcal{D}}_i\}_{u_i \in \mathcal{U}_t}$ by minimizing:

$$L = \sum_{u_i \in \mathcal{U}_t} L_i, \tag{5}$$

---

**Algorithm 1** FedKDRec

---

**Input:** Server model $\mathbf{M}_s$, clients datasets $\{\mathcal{D}_i\}_{u_i \in \mathcal{U}}$, multi-teacher KD agent $\mathbf{M}_R$, AC agent $\pi_A$ and $\mathbf{M}_C$.

**Output:** Trained server model $\mathbf{M}_s$.

server initialize model $\mathbf{M}_{s,0}$ and distributed to clients $\{\mathbf{M}_{i,0}\}_{u_i \in \mathcal{U}}$;

**for** *each round $t$* **do**

    sample a subset of clients $\mathcal{U}_t \subseteq \mathcal{U}$;

    **for** *each client $u_i \in \mathcal{U}_t$* **in parallel do**

        agent $\pi_A$ samples auxiliary dataset $\tilde{\mathcal{D}}_i$ as Sec. 3.3;

        $\mathbf{M}_{i,t+1} \leftarrow$ client $u_i$ updates client model as Eq. 3 using $\mathcal{D}_i$ and $\tilde{\mathcal{D}}_i$;

        client $u_i$ sample prediction dataset $\hat{\mathcal{D}}_i$ as Sec. 3.1;

    receive clients datasets with predictions $\{\hat{\mathcal{D}}_i\}_{u_i \in \mathcal{U}_t}$;

    generate weights $\{\omega_i\}_{u_i \in \mathcal{U}_t}$ as Sec. 3.2 using $\{\hat{\mathcal{D}}_i\}_{u_i \in \mathcal{U}_t}$;

    $\mathbf{M}_{s,t+1} \leftarrow$ update server model as Eq. 5 using $\{\omega_i\}_{u_i \in \mathcal{U}_t}$;

    optimize agent $\mathbf{M}_R$ as Sec. 3.2;

    optimize $\pi_A$ and $\mathbf{M}_C$ as Sec. 3.3 using $\{\hat{\mathcal{D}}_i\}_{u_i \in \mathcal{U}_t}$ and $\{\tilde{\mathcal{D}}_i\}_{u_i \in \mathcal{U}_t}$;

---

$$L_i = \sum_{(u_i, v_j, \overline{r}_{i,j}) \in \hat{\mathcal{D}}_i} L_{ce}(\tilde{r}_{i,j}, \overline{r}_{i,j}) = - \sum_{(u_i, v_j, \overline{r}_{i,j}) \in \hat{\mathcal{D}}_i} \overline{r}_{i,j} \log(\tilde{r}_{i,j}) + (1 - \overline{r}_{i,j}) \log(1 - \tilde{r}_{i,j}), \quad (6)$$

where $\tilde{r}_{i,j} = \mathbf{M}_{s,t}(u_i, v_j)$.

**Server-to-clients KD.** Through clients-to-server KD, the server aggregates knowledge from multiple clients and learns a stronger global model. To further enhance recommendations, the server distributes distilled knowledge back to the clients.

Algorithm 1 provides a summary of the overall procedure of FedKDRec with the incorporation of the two proposed agents.

## 3.2 CLIENTS-TO-SERVER KD: EFFECTIVE MULTI-TEACHER KD AGENT

For the clients-to-server KD phase, we formulate the integration of knowledge from multiple heterogeneous clients as a multi-teacher KD problem, where each client acts as a teacher and the server serves as the student. Accordingly, the clients' weights become a critical factor for effective distillation. Based on Eq. 2 and Eq. 5, the optimization objective is extended as:

$$L = \sum_{u_i \in \mathcal{U}_t} \omega_i L_i, \qquad (7)$$

where $\omega_i$ denotes the weight of client $u_i$. To generate the clients' weights, we design an agent to adapt to the dynamics of FR training environments quickly. The formulation is described as follows:

**State.** For each client $u_i$, we construct the state embedding $\mathbf{s}_i$ by concatenating three components:

1. Sample volume: the number of samples, reflecting the client's knowledge volume.

2. Prediction statistics: mean and variance of client outputs, characterizing their distribution.

3. Model compatibility: average loss on the server model, reflecting the client's contribution.

**Action.** The agent generates the weights as $\omega_i = softmax(\mathbf{M}_R(\mathbf{s}_i)), u_i \in \mathcal{U}_t$, where $\mathbf{M}_R$ is a neural network with a middle ReLU activation. Each weight is an action in a continuous space $(0, 1)$.

**Reward.** We define the reward as the negative loss $R_i = -L_i$ to measure the contributions to the performance of the server model, followed by normalization.

**Agent Optimizing.** We adopt the commonly used policy gradient to optimize the agent as shown in Appendix C.2. This encourages the agent to assign higher weights to clients that yield positive rewards, thereby adaptively emphasizing more informative client knowledge.

## 3.3 SERVER-TO-CLIENTS KD: EFFICIENT KD AGENT

In FedKDRec, each client is trained with an auxiliary dataset provided by the server to enhance recommendation performance. However, given the constrained communication and computational capacity on the client side, it is crucial to transfer knowledge that is both informative and lightweight, rather than relying on static heuristics. To this end, we design an actor-critic (AC) agent deployed on the server to adaptively control the sampling process. The formalization is detailed as follows:

**State.** Since the auxiliary dataset mainly consists of negative samples, the agent is designed to prioritize hard negative samples for their greater positive impact. Based on the historical client datasets $\{\hat{\mathcal{D}}_j\}_{u_j \in \mathcal{U}_{t-1}}$, we first construct an item sequence $\mathcal{V}^s$, ordered by server predictions.

For each client $u_i \in \mathcal{U}_t$, we sequentially construct a state sequence $\mathcal{S}_i = \{\mathbf{s}_{i,j}\}$, where each state embedding $\mathbf{s}_{i,j}$ for the candidate item $v_j \in \mathcal{V}^s$ is obtained by concatenating four components:

1. Prediction $\tilde{r}_{i,j}$ on $v_j$, indicating the hardness of negative items.

2. Frequency of $v_j$ in historical datasets $\{\hat{\mathcal{D}}_j\}$, reflecting its exposure in training.

3. User embedding and item embedding, providing the interaction information.

4. Number of selected samples $|\tilde{\mathcal{V}}_i|$, representing transfer cost.

To avoid excessive growth of $\mathcal{S}_i$ during the early AC training, especially when the item sequence $\mathcal{V}^s$ contains a large volume of items, we set a threshold $\mu$ as the upper bound of the size of $\mathcal{S}_i$.

**Action.** Given state embedding $\mathbf{s}_{i,j}$, the actor $\pi_A$ sequentially decides whether a candidate sample should be delivered to a client. Specifically, it outputs a probability $p_{i,j,a} = \pi_A(a_{i,j} \mid \mathbf{s}_{i,j})$ to decide the action $a_{i,j}$, where $p_{i,j,a} \in (0,1)$ and $a_{i,j} \in \{0,1\}$. $a_{i,j} = 1$ indicates that actor adds item $v_j$ into $\tilde{\mathcal{V}}_i$, while $a_{i,j} = 0$ denotes skipping. In parallel, the critic estimates the value function.

**Reward.** We design the reward to encourage the transferred knowledge to be both informative and lightweight. The informativeness of knowledge is captured by three complementary components:

1. Performance on the server model. This is evaluated as Sec. 3.2 to obtain $R_i$.

2. Item diversity. Given the item embeddings $\hat{V}_i$ of received samples from $u_i$ and $\tilde{V}_i$ of transferred samples to $u_i$, we use the KL-divergency to evaluate diversity of items as:

$$R_i^s = D_{KL}(\hat{V}_i || \tilde{V}_i). \tag{8}$$

3. Model divergency. Given the predictions $\{\overline{r}_{i,j}\}$ of client $u_i$ and $\{\tilde{r}_{i,j}\}$ of the server on the same items, we evaluate the model divergency by similarity on the predictions:

$$R_i^l = 1 - \frac{1}{|\hat{\mathcal{V}}_i|} \sum_{v_j \in \hat{\mathcal{V}}_i} cos(\overline{r}_{i,j}, \tilde{r}_{i,j}). \tag{9}$$

The overall reward $R_i^{AC}$ is computed as the average of the three normalized components.

To balance informativeness with communication efficiency, we further introduce a position-based adjustment $\frac{\mathbf{p}(s_{i,j})}{\rho|\mathcal{S}_i|}$. Earlier states are encouraged to be selected (higher reward for skipping penalties), while later states are discouraged from excessive selections. This adjustment is scaled by a factor proportional to the state's position within $\mathcal{S}_i$ (with coefficient $\rho = 4$ as Appendix C.3).

**Agent Optimizing.** We optimize the AC agent using standard policy gradient and temporal-difference error methods as Appendix C.2. Unlike conventional settings where rewards are immediately available, in our case, the reward $R_i^{AC}$ is defined over the entire state sequence $\mathcal{S}_i$. Therefore, the optimization is performed only after the client's dataset is fully received.

Table 1: Performance comparison on three datasets. Numbers in () indicate relative changes, with improvements in cyan and drops in orange, where intensity scales with the magnitude of change.

| Method | Backbone | MovieLens-100K | | MovieLens-1M | | Amazon Industrial | |
|---|---|---|---|---|---|---|---|
| | | N@10 | H@10 | N@10 | H@10 | N@10 | H@10 |
| Centralized | NeuMF | 33.16 | 63.52 | 53.23 | 77.10 | 12.09 | 20.61 |
| | NeuNCF | 32.80 | 59.07 | 47.90 | 70.81 | 10.28 | 18.28 |
| | LightGCN | 40.44 | 65.96 | 52.13 | 76.62 | 18.49 | 30.20 |
| Parameter-transfer FR | FedMF | 34.55(+4.21) | 56.14(-11.62) | 24.37(-54.22) | 43.14(-44.05) | 6.70(-44.52) | 12.62(-38.77) |
| | FedNCF | 36.19(+10.32) | 58.45(-1.04) | 35.63(-25.62) | 58.78(-16.98) | 9.39(-8.58) | 16.51(-9.64) |
| | FedPerGNN | 33.38(-17.46) | 55.08(-16.50) | 32.33(-37.98) | 58.91(-23.12) | 8.60(-53.48) | 15.00(-50.33) |
| PTF-FedRec | NeuMF | 36.00(+8.56) | 64.28(+1.20) | 44.35(-16.69) | 71.34(-7.47) | 11.29(-6.61) | 20.18(-2.08) |
| | NeuNCF | 35.76(+9.02) | 60.83(+2.98) | 45.87(-4.25) | 68.66(-3.03) | 10.05(-2.24) | 18.10(-0.96) |
| | LightGCN | 40.75(+0.78) | 64.84(-1.70) | 51.96(-0.32) | 75.22(-1.83) | 11.00(-40.49) | 19.05(-36.90) |
| FedKDRec | NeuMF | 40.02(+20.68) | 67.13(+5.68) | 53.63(+0.75) | 76.90(-0.26) | 12.65(+4.70) | 21.67(+5.15) |
| | NeuNCF | 36.78(+12.12) | 64.34(+8.92) | 48.78(+1.83) | 72.32(+2.13) | 10.49(+2.07) | 18.88(+3.31) |
| | LightGCN | 42.35(+4.73) | 66.62(+1.00) | 53.47(+2.57) | 77.83(+1.58) | 19.02(+2.88) | 31.88(+5.59) |

## 4 EXPERIMENTS

### 4.1 EXPERIMENT SETUP

**Datasets.** Three public recommendation datasets are used: MovieLens-100K (Harper & Konstan, 2016), MovieLens-1M (Harper & Konstan, 2016), Amazon Industrial (Ni et al., 2019).

**Baselines.** We compare our proposed FedKDRec with three groups of baselines:

(i) Centralized recommendation baselines: NeuMF (He et al., 2017), NeuNCF (He et al., 2017), and LightGCN (He et al., 2020), which represent strong backbones trained with centralized data.

(ii) Parameter-transfer FR baselines: FedMF (Chai et al., 2021), FedNCF (Perifanis & Efraimidis, 2022), and FedPerGNN Wu et al. (2022), which follow the parameter-sharing paradigm in FR.

(iii) KD based FR baseline: PTF-FedRec (Yuan et al., 2024), which leverages prediction transfer instead of parameter exchange and confidence based sampling to select hard samples for clients.

**Evaluation Metrics.** We adopt the widely used top-K recommendation evaluation metrics: hit ratio (H@K) and normalized discounted cumulative gain (N@K). Both metrics reflect the ranking quality, where higher values indicate better performance.

### 4.2 PERFORMANCE COMPARISONS

**Overall Performance.** We evaluate our proposed FedKDRec with baselines, and the results are summarized in Table 1. Several observations can be drawn. First, KD-based FR methods (PTF-FedRec and FedKDRec) outperform parameter-transfer FR methods under the same backbones. This highlights the advantage of transferring prediction knowledge instead of model parameters.

Second, FedKDRec consistently outperforms all baselines, including centralized recommendation methods, across different datasets. This superiority improvement comes from two key factors: 1) FedKDRec integrates both hard labels and soft predictions, thus combining ground-truth supervision with additional knowledge distilled from client models; 2) FedKDRec employs a multi-teacher KD strategy that adaptively assigns weights to client knowledge, enabling more effective aggregation under heterogeneous client models and diverse local data distributions.

Third, FedKDRec brings substantial improvements in certain settings compared to PTF-FedRec. For example, on MovieLens-1M, FedKDRec(NeuMF) achieves 20.9% and 7.8% improvements in N@10 and H@10 compared with PTF-FedRec(NeuMF). On Amazon Industrial, Fed-KDRec(LightGCN) achieves even larger gains, with 72.9% and 67.3% improvements in N@10 and H@10 over PTF-FedRec(LightGCN). These results clearly demonstrate the effectiveness of our agent-enhanced KD framework in boosting both ranking quality and coverage.

Table 2: Ablation study of different variants on three datasets. Numbers in () indicate the relative performance drop (%), where color scales with the magnitude of change. Best results are in **bold**.

| Backbone | Variant | MovieLens-100K | | MovieLens-1M | | Amazon Industrial | |
|---|---|---|---|---|---|---|---|
| | | N@10 | H@10 | N@10 | H@10 | N@10 | H@10 |
| NeuMF | -m-c~s | 36.00(-10.04) | 64.28(-4.23) | 44.35(-17.31) | 71.34(-7.23) | 11.29(-10.80) | 20.18(-6.88) |
| | -m | 36.86(-7.88) | 64.90(-3.32) | 44.35(-17.31) | 74.81(-2.73) | 11.99(-5.29) | 20.51(-5.32) |
| | -c | 36.62(-8.49) | 64.23(-4.31) | 48.86(-8.90) | 74.45(-3.19) | 11.23(-11.23) | 20.71(-4.43) |
| | -s | 35.67(-10.86) | 63.92(-4.77) | 52.00(-3.04) | 76.42(-0.63) | 11.96(-5.47) | 20.42(-5.76) |
| | ~s | 39.91(-0.26) | 66.60(-0.79) | 52.86(-1.43) | 76.33(-0.74) | 12.10(-4.41) | 20.60(-4.92) |
| | FedKDRec | **40.02** | **67.13** | **53.63** | **76.90** | **12.65** | **21.67** |
| NeuNCF | -m-c~s | 35.76(-2.77) | 60.83(-5.46) | 45.87(-5.97) | 68.66(-5.05) | 10.05(-4.22) | 18.10(-4.13) |
| | -m | 36.17(-1.65) | 64.05(-0.45) | 48.05(-1.50) | 68.08(-5.86) | 10.26(-2.21) | 18.52(-1.94) |
| | -c | 35.96(-2.22) | 61.32(-4.68) | 47.32(-3.01) | 70.39(-2.66) | 10.19(-2.89) | 18.52(-1.91) |
| | -s | 34.69(-5.66) | 60.55(-5.88) | 48.04(-1.51) | 71.39(-1.28) | 10.38(-1.01) | 18.66(-1.20) |
| | ~s | 36.57(-0.55) | 62.40(-3.02) | 48.26(-1.07) | 71.98(-0.47) | 10.32(-1.57) | 18.57(-1.65) |
| | FedKDRec | **36.78** | **64.34** | **48.78** | **72.32** | **10.49** | **18.88** |
| LightGCN | -m-c~s | 40.75(-3.77) | 64.84(-2.67) | 51.96(-2.81) | 75.22(-3.36) | 11.00(-42.15) | 19.05(-40.24) |
| | -m | 41.18(-2.76) | 65.02(-2.40) | 52.25(-2.27) | 75.46(-3.04) | 11.63(-38.84) | 19.78(-37.97) |
| | -c | 41.29(-2.51) | 64.85(-2.65) | 52.29(-2.20) | 76.15(-2.16) | 18.64(-2.01) | 31.01(-2.73) |
| | -s | 40.25(-4.96) | 65.22(-2.10) | 52.28(-2.22) | 77.00(-1.07) | 17.12(-9.99) | 28.63(-10.21) |
| | ~s | 40.55(-4.25) | 65.71(-1.37) | 53.04(-0.79) | 77.78(-0.07) | 18.77(-1.31) | 31.62(-0.82) |
| | FedKDRec | **42.35** | **66.62** | **53.47** | **77.83** | **19.02** | **31.88** |

**Ablation Study.** We conduct ablation to evaluate the contribution of the mixing strategy (Eq. 4), the client-to-server multi-teacher KD agent (Sec. 3.2), and the server-to-client AC agent (Sec. 3.3). Table 2 summarizes the results across three backbones and datasets. Our findings are as follows:

(i) Effect of mixing strategy ($-m$). Removing the mixing of hard labels and soft predictions leads to a consistent performance drop across backbones, particularly in LightGCN where the degradation reaches $-38.84\%$ in N@10 on Industrial. This shows that combining hard with soft labels provides a better balance between supervision and teacher model information.

(ii) Effect of clients-to-server agent ($-c$). Disabling the client-to-server agent consistently reduces performance, especially with the degradation on the Amazon Industrial (e.g., $-11.23\%$ on N@10 for NeuMF). This validates the insight that clients contribute knowledge of varying quality, and static aggregation is vulnerable. By leveraging richer state information, our agent adaptively emphasizes reliable clients, making dynamic weighting crucial for robust knowledge transfer.

(iii) Effect of server-to-clients agent ($\sim s$ vs. $-s$). Replacing our server-to-client agent with the confidence-based selection strategy in PTF-FedRec ($\sim s$) still maintains reasonable performance, but completely removing this phase ($-s$) causes substantial degradation, particularly on LightGCN. This confirms that server-to-client knowledge transfer is effective, and also highlights the limitation of static confidence-based heuristics. By modeling the selection as a sequential decision-making process, our agent adaptively balances informativeness and efficiency.

(iv) Combined removal ($-m-c\sim s$). When all three components are removed, the performance decreases most dramatically. This verifies that the proposed modules, mixing, client-to-server KD, and server-to-client KD, play vital roles. In summary, these results demonstrate that each proposed component contributes meaningfully to FedKDRec, and their combination provides robust improvements across architectures and datasets.

## 4.3 COMMUNICATION COST

We further evaluate the communication efficiency of FedKDRec compared with existing FR baselines during the phase where clients receive models or predictions. As the average per-client communication cost per round shown in Table 3, FedKDRec achieves a substantial reduction in communication cost relative to parameter-transfer FR methods, ow-

Table 3: Average per-client communication cost.

| Method | ML-100K | ML-1M | Amazon-Ind. |
|---|---|---|---|
| FedMF | 0.205 MB | 0.482 MB | 0.651 MB |
| FedNCF | 0.166 MB | 0.305 MB | 0.389 MB |
| FedPerGNN | 0.103 MB | 0.241 MB | 0.326 MB |
| PTF-FedRec | 4.578 KB | 4.578 KB | 4.578 KB |
| FedKDRec(NeuMF) | 0.175 KB | 0.169 KB | 0.037 KB |
| FedKDRec(NeuCF) | 0.070 KB | 0.175 KB | 0.089 KB |
| FedKDRec(LightGCN) | 0.084 KB | 0.172 KB | 0.037 KB |

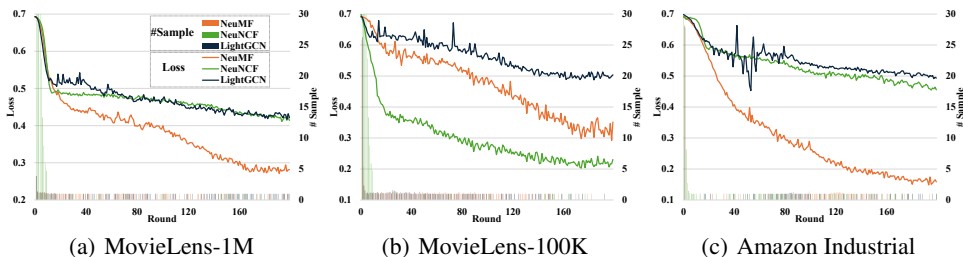

|  |  |  |
|:---:|:---:|:---:|
| (a) MovieLens-1M | (b) MovieLens-100K | (c) Amazon Industrial |

Figure 2: Communication cost and training loss per round. Bars indicate the average number of transferred predictions, while the line denotes the server-side training loss.

Table 4: Ablation on agent design with NeuNCF backbone.

| Agent | Variant | MovieLens-100K | | | MovieLens-1M | | |
|---|---|---|---|---|---|---|---|
| | | N@10 | H@10 | Comm. (KB) | N@10 | H@10 | Comm. (KB) |
| Clients-to-server | w/o sample size | 35.45 | 62.99 | – | 47.62 | 70.63 | – |
| | w/o statistics info | 35.66 | 62.98 | – | 47.60 | 70.73 | – |
| | w/o BCE loss | 35.91 | 61.40 | – | 47.99 | 70.14 | – |
| Server-to-clients | w/o $R_i$ | 36.25 | 63.32 | 0.228 | 47.69 | 71.46 | 3.257 |
| | w/o $R_i^s$ | 36.22 | 63.69 | 0.184 | 47.75 | 72.20 | 2.304 |
| | w/o $R_i^l$ | 36.16 | 63.26 | 0.268 | 48.28 | 71.85 | 3.797 |
| FedKDRec | | **36.78** | **64.34** | **0.070** | **48.78** | **72.32** | **0.175** |

ing to its design of transferring only distilled knowledge rather than full model parameters. Even when compared with the communication-efficient PTF-FedRec based on federated KD, FedKDRec consistently exhibits lower overhead across all datasets.

In addition, Figure 2 provides further insights into the communication behavior of FedKDRec. We can observe that the communication cost decreases sharply during the early training rounds, coinciding with the rapid drop in loss, as the AC agent learns to select fewer but more informative samples for transfer. This demonstrates that FedKDRec not only reduces communication overhead but also adaptively balances efficiency and effectiveness as the server model converges.

### 4.4 INSIGHT OF AGENTS

We conduct ablation studies on the two proposed agents to examine the effect of different state components for the client-to-server agent and reward design for the server-to-client agent. The results on the NeuNCF backbone are shown in Table 4, from which several insights can be drawn:

(i) Full FedKDRec consistently outperforms all ablated variants, achieving the highest accuracy while reducing communication cost. Notably, the communication reduction is more prominent in larger datasets (MovieLens-1M), where the volume of transferred predictions is higher.

(ii) Clients-to-server agent (state design). Removing any single component (sample size, statistical information, or BCE loss) leads to a noticeable performance drop. In particular, excluding the BCE loss results in the largest degradation (e.g., $-4.57\%$ on H@10 in MovieLens-100K), indicating that local training quality is a crucial signal for weighting client contributions. This highlights the importance of integrating both data- and model-related statistics in the state representation.

(iii) Server-to-clients agent (reward design). The reward components ($R_i$, $R_i^s$, $R_i^l$) jointly balance effectiveness and efficiency. Specifically, $R_i$ reflects performance on the server model, ensuring cost-aware improvements; $R_i^s$ captures sample diversity, preventing the agent from overfitting to redundant subsets and stabilizing knowledge transfer; and $R_i^l$ measures model discrepancy, encouraging the agent to prioritize informative samples while suppressing redundant communication.

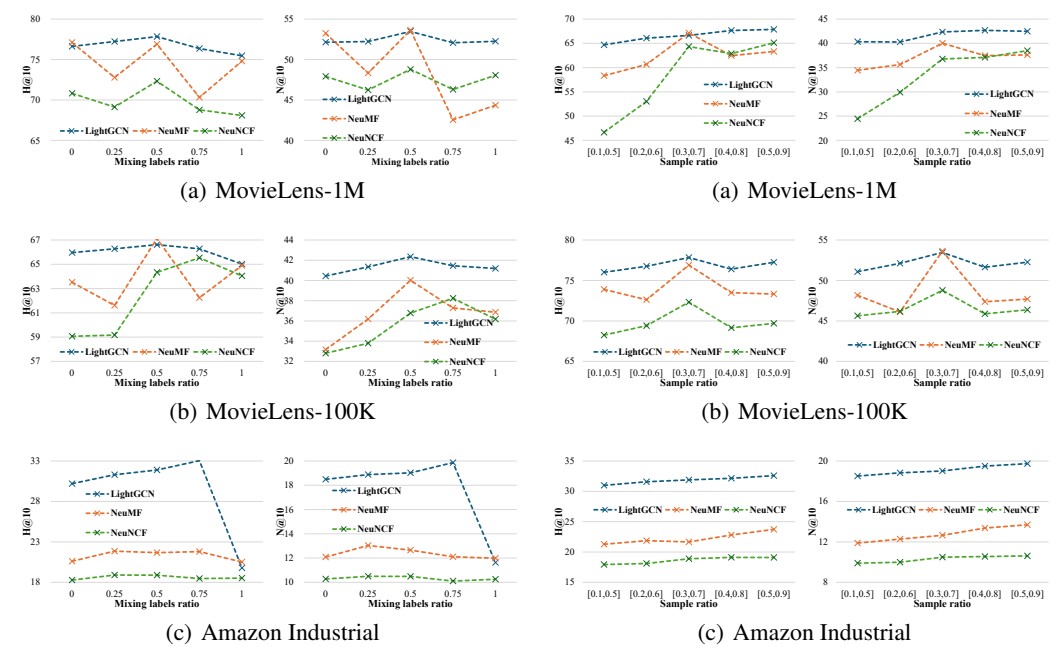

(a) MovieLens-1M        (a) MovieLens-1M

(b) MovieLens-100K      (b) MovieLens-100K

(c) Amazon Industrial      (c) Amazon Industrial

Figure 3: Hyperparameter study on $\alpha$.     Figure 4: Hyperparameter study on $\beta$.

### 4.5 HYPERPARAMETERS SENSITIVITY

We evaluate two hyperparameters: mixing labels ratio $\alpha$ and sample ratio $\beta$.

From the results on $\alpha$ in Figure 3, several observations can be made: 1) For LightGCN, the best results are consistently achieved when mixing hard labels and predictions, with the peak performance observed at $\alpha = 0.5$. In contrast, on the Industrial dataset, its performance drops significantly when $\alpha = 1.0$ (w/o mixing). 2) On the MovieLens-1M dataset, the performance of NeuMF and NeuNCF shows instability around $\alpha = 0.5$. Interestingly, these models perform better at the extremes, i.e., $\alpha = 0$ (only hard labels) and $\alpha = 1.0$ (only predictions), compared with intermediate values such as $\alpha = 0.25$ or $\alpha = 0.75$. 3) Overall, setting $\alpha = 0.5$, which assigns equal weights to hard labels and predictions, tends to provide a robust trade-off and yields strong performance in many cases.

Figure 4 reports the results on $\beta$. Several insights can be drawn from the results: 1) For the Industrial and MovieLens-1M datasets, performance improves overall as $\beta$ increases, which is consistent with the intuition that a larger transfer ratio enables the server to access more informative knowledge. 2) On the MovieLens-100K dataset, the best results are achieved when $\beta$ lies in the range $[0.3, 0.7]$. Notably, when $\beta$ increases further into $[0.4, 0.8]$, the performance drops significantly, even lower than that of a smaller ratio such as $[0.2, 0.6]$ with the LightGCN backbone. This suggests that in relatively small datasets, excessive transfer may introduce noise and lead to overfitting. 3) Furthermore, under the favorable range $\beta \in [0.3, 0.7]$, the NeuMF backbone consistently achieves the best results on both MovieLens-1M and MovieLens-100K compared with other values of $\beta$.

## 5 CONCLUSION

In this paper, we propose FedKDRec, a federated recommendation framework that integrates bidirectional knowledge distillation with reinforcement learning agents to simultaneously enhance privacy protection, communication efficiency, and recommendation accuracy. Instead of exchanging raw parameters that risk model leakage and incur high communication cost, FedKDRec transfers mixing soft predictions under a privacy-preserving mechanism. To further enhance effectiveness, a server-oriented RL agent adaptively reweights heterogeneous client knowledge in multi-teacher distillation, while a client-oriented actor-critic agent selectively delivers informative yet lightweight samples to clients. Extensive experiments confirm that FedKDRec outperforms the baselines.

## 6 ETHICS STATEMENT

This work does not involve human subjects, sensitive personal data, or any proprietary datasets. All datasets used in this study are publicly available and commonly used in prior research. We have taken care to ensure that our methods and results do not raise safety, privacy, or fairness concerns.

## 7 REPRODUCIBILITY STATEMENT

We set random seeds to ensure reproducibility and report the average performance over five independent runs. The source code is uploaded on **Supplementary Materials** with the used datasets, requirements file, and experimental settings including random seeds.

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

## A    GenAI Usage Disclosure

Generative AI tools were used solely to improve the quality of the text. No generative AI tools were used in any other stages of the research, including the design, implementation, analysis, or generation of code, data, or experimental results.

## B    Related Work

**Federated Recommendation (FR).** FR have recently attracted growing attention as it enables collaborative model training across distributed devices while preserving user privacy. Early works such as FedMF (Chai et al., 2021), FedNCF (Perifanis & Efraimidis, 2022), and FedPerGNN (Wu et al., 2022) extend classical recommendation models into federated settings by adopting parameter-sharing paradigms. While effective, these approaches often incur high communication costs and suffer from data heterogeneity issues. To address these challenges, more recent research has proposed frameworks tailored to the unique constraints of FRS. For example, NodeRec+ (O'Reilly-Morgan et al., 2025) provides a flexible and extensible FL framework for recommendation, supporting multiple topologies and enabling rapid prototyping and benchmarking in decentralized settings. LightFR (Zhang et al., 2023b) focuses on efficiency and privacy by combining federated learning with hashing techniques, generating compact binary codes for large-scale recommendation while defending against gradient leakage attacks. PrivFR (Zhang et al., 2025) further advances efficiency by introducing shared hash embeddings, which not only reduce storage and communication overhead but also address the out-of-vocabulary (OOV) issue, a practical challenge often overlooked in prior studies. Beyond these systems-level innovations, another line of work explores knowledge distillation as an alternative to direct parameter sharing, e.g., PTF-FedRec (Yuan et al., 2024), which transfers predictions instead of gradients or parameters to improve personalization under heterogeneous data distributions. Compared to the above designs, KD-based FRS methods offer greater flexibility in model heterogeneity and privacy preservation.

**Knowledge Distillation (KD) in Federated Learning (FL).** KD has recently been widely explored in FL as an alternative to parameter-transfer schemes, aiming to alleviate issues such as model homogeneity, high communication cost, and poor performance under heterogeneous data distributions. Mora et al. (2024) provide a systematic review and classification of KD-based FL algorithms, highlighting their trade-offs compared to traditional FedAvg-like approaches. Beyond general surveys, a line of work focuses on adapting KD to specific FL challenges. For instance, Ma et al. (2022) introduce Continual Federated Distillation (CFeD) to mitigate catastrophic forgetting in non-stationary data distributions by performing KD on both clients and the server, with different clients specializing in new versus old tasks. In the vertical FL setting, Huang et al. (2023) propose VFedTrans, which leverages representation distillation to enable knowledge transfer across healthcare institutions without directly sharing sensitive features, thus broadening the applicability of VFL to heterogeneous collaboration networks. To handle heterogeneous data and model architectures, Yang et al. (2023) design FedACK, a GAN-based adversarial and contrastive KD framework that maintains consistent feature spaces across clients while improving communication efficiency in cross-lingual social bot detection. From the perspective of personalization, Chen et al. (2023) propose FedHKD, where clients exchange hyper-knowledge (soft predictions and representation statistics) with the server, enabling improved global and personalized model performance without requiring public datasets or generative models. Finally, Afonin & Karimireddy (2022) provide a theoretical analysis of KD in FL through a kernel regression framework, revealing fundamental limitations of distillation under data heterogeneity while offering new protocol designs consistent with empirical results.

## C  METHODOLOGY

### C.1  PROOF ON EQUIVALENCE OF MIXED LABELS

We show that training with the mixed label in Eq. 6 is equivalent to the KD-style loss in Eq. 1. For a single pair $(u_i, v_j)$, let the student prediction be $\tilde{r}$, the ground-truth label $r$, and the teacher soft prediction $\hat{r}$. Define the mixed label as Eq. 4:

$$\bar{r} = \alpha r + (1 - \alpha)\hat{r}.$$

Substituting $\bar{r}$ into the binary cross-entropy (BCE) loss yields

$$\begin{aligned}
\mathcal{L}(\tilde{r}, \bar{r}) &= -\bar{r}\log\tilde{r} - (1 - \bar{r})\log(1 - \tilde{r}) \\
&= -\Big[(\alpha r + (1-\alpha)\hat{r})\log\tilde{r} + (1 - \alpha r - (1-\alpha)\hat{r})\log(1-\tilde{r})\Big] \\
&= \alpha\big[-r\log\tilde{r} - (1-r)\log(1-\tilde{r})\big] + (1-\alpha)\big[-\hat{r}\log\tilde{r} - (1-\hat{r})\log(1-\tilde{r})\big] \\
&= \alpha\,\mathcal{L}_{ce}(\tilde{r}, r) + (1-\alpha)\,\mathcal{L}_{ce}(\tilde{r}, \hat{r}).
\end{aligned}$$

Extending to the entire set $\hat{\mathcal{D}}_i$ yields

$$L_i = \alpha\sum\mathcal{L}_{ce}(\tilde{r}_{i,j}, r_{i,j}) + (1-\alpha)\sum\mathcal{L}_{ce}(\tilde{r}_{i,j}, \hat{r}_{i,j}).$$

This shows that minimizing BCE with the mixed label $\bar{r}$ is equivalent to the KD-style objective

$$L_{KD} = \alpha\,\mathcal{L}_{ce}(\tilde{r}^S, r) + (1-\alpha)\,D(\tilde{r}^S, \hat{r}^T),$$

when $D$ is chosen as cross-entropy. If instead $D$ is chosen as the KL divergence, note that

$$D_{KL}(\hat{r}\,\|\,\tilde{r}) = \mathcal{L}_{ce}(\tilde{r}, \hat{r}) - H(\hat{r}),$$

where $H(\hat{r})$ is the entropy of the teacher distribution and independent of $\tilde{r}$. Thus, the two losses differ only by a constant term and are therefore equivalent in optimization.

This equivalence provides a theoretical justification for using mixed labels as a unification of hard-label supervision and teacher-guided knowledge distillation.

### C.2  OPTIMIZATION ON AGENT

We adopt the commonly used policy gradient to optimize the clients-to-server KD agent by minimizing the objective as:

$$-\sum_{u_i \in \mathcal{U}_t} R_i \log(\omega_i). \tag{10}$$

This encourages the agent to assign higher weights to clients that yield positive rewards, thereby adaptively emphasizing more informative client knowledge.

Formally, the critic is trained to minimize the mean squared TD error:

$$\mathcal{L}_C = \frac{1}{|\mathcal{S}_i|}\sum_{s_{i,j}\in\mathcal{S}_i}\big(R_i^{AC} + \gamma V(s_{i,j+1}) - V(s_{i,j})\big)^2, \tag{11}$$

where $V(\cdot)$ is the critic's value function and $\gamma$ is the discount factor.

The actor is optimized by maximizing the expected cumulative reward through policy gradient:

$$\mathcal{L}_A = -\frac{1}{|\mathcal{S}_i|}\sum_{s_{i,j}\in\mathcal{S}_i}\log\pi_A(a_{i,j}\mid s_{i,j})\cdot A(s_{i,j}, a_{i,j}), \tag{12}$$

where $A(s_{i,j}, a_{i,j}) = R_i^{AC} + \gamma V(s_{i,j+1}) - V(s_{i,j})$ is the advantage function estimated by the critic.

## C.3 REWARD ADJUSTMENT

To balance informativeness with communication efficiency, we further adjust the reward $R_i^{AC}$ at the state level. Specifically, we encourage more aggressive sampling at early stages and more conservative behavior later, while penalizing excessive selections. For a state $s_{i,j} \in \mathcal{S}_i$ with position $\mathbf{p}(s_{i,j})$, the adjustment factor is $\frac{\mathbf{p}(s_{i,j})}{\rho|\mathcal{S}_i|}$, where $\rho = 4$. If $a_{i,j} = 1$ (sample selected), the adjusted reward is:

$$R_i^{AC} - \frac{\mathbf{p}(s_{i,j})}{\rho|\mathcal{S}_i|}\|R_i^{AC}\|_1.$$

If $a_{i,j} = 0$ (sample skipped), the adjusted reward is:

$$R_i^{AC} + \frac{\mathbf{p}(s_{i,j})}{\rho|\mathcal{S}_i|}\|R_i^{AC}\|_1.$$

# D EXPERIMENTS

## D.1 DATASETS

Three public recommendation datasets are used: MovieLens-100K (Harper & Konstan, 2016), MovieLens-1M (Harper & Konstan, 2016), Amazon Industrial (Ni et al., 2019). The statistics of these datasets are summarized in Table 5.

Table 5: Dataset statistics.

| Dataset | # Users | # Items | # Interactions | Sparsity |
|---|---|---|---|---|
| ML100K | 943 | 1,682 | 100,000 | 93.70% |
| ML1M | 6,040 | 3,706 | 1,000,209 | 95.53% |
| Industrial | 11,041 | 5,334 | 77,071 | 99.87% |

## D.2 BASELINES

We conduct experiments on three groups of baselines:

1) *Centralized recommendation baselines.*

**NeuMF** (He et al., 2017): A hybrid neural recommendation model that combines generalized matrix factorization (GMF) and multi-layer perceptron (MLP) to capture both linear and non-linear user–item interactions.

**NeuNCF** (He et al., 2017): A neural collaborative filtering variant that replaces the element-wise product in GMF with a neural network, thus enhancing the expressiveness of interaction modeling.

**LightGCN** (He et al., 2020): A lightweight graph convolutional network designed for recommendation, which propagates user–item embeddings on the interaction graph without non-linear transformations or feature transformations, making it efficient and effective.

2) *Parameter-transfer federated recommendation baselines.*

**FedMF** (Chai et al., 2021): A federated adaptation of matrix factorization, where clients train local user embeddings and the server maintains global item embeddings. The parameter synchronization between the server and clients enables collaborative learning but incurs communication overhead.

**FedNCF** (Perifanis & Efraimidis, 2022): A federated extension of neural collaborative filtering, where model parameters are trained locally on each client and aggregated on the server. This approach enhances model expressiveness compared to FedMF but is more vulnerable to data heterogeneity across clients.

**FedPerGNN** Wu et al. (2022): A personalized federated recommendation method based on graph neural networks (GNNs). It decouples parameters into global and personalized parts, where global

parameters are aggregated while personalized parameters are retained locally, aiming to balance personalization and knowledge sharing.

3) *Knowledge-distillation based federated recommendation baseline.*

**PTF-FedRec** (Yuan et al., 2024): A recent KD-based FR framework that avoids parameter transfer and instead relies on transferring predictions. Specifically, the server selects a subset of items using a confidence-based hard sampling strategy and sends predictions to clients. This design significantly reduces communication costs compared to parameter-transfer methods while preserving recommendation accuracy. However, its sample selection strategy is relatively static and may not fully exploit the informativeness of transferred knowledge.

### D.3 IMPLEMENTATION DETAILS

In our FR setting, each client is locally trained for 2 epochs, while the server is trained for 4 epochs per round. The client participation ratio is fixed at $\mathcal{C} = 10\%$, and the total number of federated training rounds is 200. For all models, we set the embedding dimension as 32. For FedKDRec, we set the mixing label ratio $\alpha = 0.5$, the client sampling ratio $\beta$ is randomly chosen from $[0.3, 0.7]$, and the swap ratio $\lambda = 0.1$. For the client sampling agent, the maximum number of transferred samples per client is capped at $\mu = 30$.

To ensure fair comparisons, we carefully configure the backbone architectures across datasets. For the NeuNCF backbone, we use a three-layer structure $[64, 128, 64]$ on the MovieLens-1M and Amazon Industrial datasets, while a two-layer structure $[128, 64]$ is adopted on MovieLens-100K due to its smaller scale. For LightGCN, we follow the original design with 3 propagation layers.

For all methods, item embeddings are initialized using an autoencoder trained on side information. In addition, we leverage the pre-trained language model Sentence-T5 (Ni et al., 2022) to encode textual item attributes (e.g., titles and descriptions), which are then fused into the initial embeddings. This initialization strategy ensures that models can exploit both interaction data and content semantics.

We employ the Adam optimizer for all experiments, with the learning rate set in $\{1e{-}3, 5e{-}3, 1e{-}2\}$ and weight decay of $\{1e{-}5, 1e{-}6, 1e{-}7, 1e{-}8\}$. The best configuration is reported.

Our implementation is built on the open-source library FuxiCTR (Zhu et al., 2021), which provides unified implementations of various recommendation models. We implement our framework in Py-Torch, and all experiments are conducted with NVIDIA GeForce RTX 4090 GPUs. We set random seeds to ensure reproducibility and report the average performance over five independent runs.

