# OpenReview forum: "Federated Recommendation with Reinforcement Learning based Knowledge Distillation"
_ICLR.cc/2026/Conference — Submitted to ICLR 2026_

### Official Review · Reviewer_eQK2 · 2025-10-27

**Soundness:** 3
**Presentation:** 4
**Contribution:** 2
**Rating:** 4
**Confidence:** 3

**Summary:**

This paper proposes FedKDRec, a privacy-preserving and communication-efficient federated recommendation (FR) framework. FedKDRec employs a bidirectional knowledge distillation (KD) that allows clients and the server to exchange soft predictions, guided by two reinforcement learning (RL) agents, namely a server-side agent adaptively weights clients and a client-side agent selects informative samples for knowledge transfer. Experiments seem to show advantageous performances compared to multiple categories of baselines.

**Strengths:**

1. Privacy and communication are among top issues faced by modern FedRec systems

2. It might be the first work to integrate RL for adaptive bidirectional KD weighting in a federated recommendation context.

3. Experiments show consistent gains across datasets and baselines.

**Weaknesses:**

1. The experiments lack comparisons with more works published in recent venues, only one of the current baselines is from 2024 while the others are all before 2022, which severely damages the credibility of the test.

2. The paper lacks theoretical analysis or experiments for privacy protection analysis. Some math analysis of privacy robustness or a quantification test against common attack schemes would be helpful.

3. Runtime and sample-efficiency costs of training two RL agents are not analyzed. The stability of the RL agents under non-stationary FR data is also left for exploration.

**Questions:**

Please check comments in weakness. Other questions include:

1. Can the authors provide stability analysis of the RL policies across datasets?

2. Can the authors provide privacy guarantee proof via theoretical analysis or/and experiements? Besides, could privacy be improved via differential privacy?

3.  Given RL normally require continue incremental training when dataset varies along the time, does the cost of RL training impact the overall efficiency?

---

### Official Review · Reviewer_RyQL · 2025-10-27

**Soundness:** 2
**Presentation:** 1
**Contribution:** 2
**Rating:** 2
**Confidence:** 3

**Summary:**

This paper proposes FedKDRec, a federated recommendation framework that uses bidirectional knowledge distillation (KD) between clients and server, enhanced by two RL-based agents: (1) a server-oriented agent for adaptive multi-teacher client weighting, and (2) a client-oriented actor-critic agent for selective sample transfer. While addressing relevant challenges in federated recommendation, the work suffers from incremental novelty, weak literature positioning, and limited experimental validation that undermines its contributions.

**Strengths:**

- **Comprehensive experimental scope**: Evaluates on three datasets (MovieLens-100K, MovieLens-1M, Amazon Industrial) with three backbone architectures (NeuMF, NeuNCF, LightGCN) and detailed ablation studies
- **Practical agent design**: The two-agent framework (multi-teacher weighting + adaptive sampling) is well-motivated for addressing heterogeneity and communication efficiency
- **Strong communication efficiency**: Achieves 26-96× reduction compared to parameter-transfer methods (Table 3), and outperforms PTF-FedRec baseline

**Weaknesses:**

**Critical:**


1. **Claimed novelty is overstated**. The abstract claims FedKDRec is "novel" , but:
   - Bidirectional KD in FL is not new: FedKD (Wu et al. Nature Comm. 2022) already does adaptive mutual distillation between mentor/mentee
   - Multi-teacher KD for FL is well-established: Multiple works (2022-2025) use multi-teacher distillation
   - Mixed-label distillation (hard + soft) is standard practice in KD
   - RL-based teacher weighting exists: Yang et al. AAAI 2025 explicitly does "Multi-teacher Knowledge Distillation with Reinforcement Learning"
   - The actual novelty is applying these existing techniques to federated recommendation as an application, which is an engineering contribution, not a fundamental methodological advance.

2. **Missing critical baselines**. The paper compares against only one KD-based FR baseline (PTF-FedRec) but omits some seminar works. Without these comparisons, claims of superiority over "existing KD-based baselines"  are unsubstantiated.


**Significant issues:**


3. **Method inconsistency with equivalence claim**. Appendix C.1 "proves" that mixed-label BCE is equivalent to KD loss when $D$ is cross-entropy. Howeve the main paper uses CE for $D$ but never justifies why KL divergence (standard in KD literature) isn't used


4. **Agent design lacks principled justification**. Both agents use RL formulations but:
   - State design: Why these specific features? No ablation on alternative state representations (e.g., gradient norms, loss trajectories)
   - Reward design: The server-to-client reward combines three components (R_i, R^s_i, R^l_i) with equal weighting. Why equal? No ablation on different weightings
   - Position-based adjustment: The formula $p(s_{i,j})/(ρ|S_i|)$ with $ρ=4$ is arbitrary. Appendix C.3 provides the formula but no justification for $ρ=4$
   - Baseline comparison: Why not compare against simpler non-RL heuristics (e.g., uncertainty sampling, diversity-based selection)?

5. **Experimental limitations**:
   - Limited client heterogeneity: All experiments use 10% participation ratio and fixed data splits. No evaluation on varying heterogeneity levels (e.g., extreme skew, concept drift)
   - No cold-start analysis: Federated recommendation critically needs cold-start handling, but experiments only report overall metrics
   - Statistical significance: Table 1 shows improvements but no confidence intervals or significance tests. Given five runs, where are error bars?

6. **Privacy claims are weak**. The paper claims "privacy-preserving mechanism"  but:
   - Sample+mix+swap strategies provide $(ϵ, δ)$-DP "as demonstrated in Sun & Lyu 2021" , but no privacy analysis is provided in this paper
   - No evaluation of privacy-utility tradeoff under different privacy budgets
   - Soft predictions can still leak information (model inversion attacks) - no discussion
   - The swapping mechanism (λ=0.1) exchanges labels of positive/negative samples  - how does this affect model quality? No analysis

7. **Results interpretation issues**:
   - Centralized baseline: FedKDRec sometimes outperforms centralized training (Table 1) , which is suspicious. Federated settings should not systematically beat centralized due to less data. This suggests either: (a) centralized baseline is poorly tuned, or (b) autoencoder initialization gives unfair advantage
   - Parameter-transfer FR baselines underperform badly: FedMF, FedNCF, FedPerGNN show massive drops (-54% to -17%). This is inconsistent with prior literature showing these methods work reasonably well. Suggests unfair experimental setup or poor hyperparameter tuning

8. **Hyperparameter sensitivity reveals instability**. Figure 3 shows:
   - NeuMF/NeuNCF perform better at extremes (α=0, α=1) than middle values , contradicting the claim that mixing is beneficial
   - LightGCN drops dramatically at α=1 on Industrial (-38.84%) , but authors set α=0.5 for all experiments without per-model tuning
   - This suggests the method is highly sensitive to α and requires careful per-task tuning

**Minor Issues:**

9. **Writing quality needs improvement**: For example, "Thus enabling clients to collaboratively train a global model" (missing subject), "Extensive experiments across diverse datasets and models demonstrate that FedKDRec **significantly achieves**" (awkward phrasing)

10. **Notation inconsistency**: Uses $r$ for both hard labels and mixed labels in Section 3.1 , making it confusing

**Questions:**

1. **Why does FedKDRec outperform centralized training?** Table 1 shows this repeatedly. Can you provide a principled explanation or investigate if experimental setup favors federated methods?

2. **Statistical significance?** Please add error bars and significance tests to Table 1. Which improvements are actually significant at p<0.05?

3. **Why do parameter-transfer baselines perform so poorly?** FedMF/FedNCF/FedPerGNN show massive drops inconsistent with prior work. Can you verify your implementations and hyperparameter tuning?


4. **Sensitivity to hyperparameters?** Figure 3 shows extreme sensitivity to $\alpha$. How do you choose $\alpha$ in practice without validation data?


5. **Why use CE instead of KL for knowledge distillation?** Standard KD uses KL divergence with temperature. Your equivalence proof assumes fixed teacher entropy, which doesn't hold in federated settings.

---

### Official Review · Reviewer_3Cof · 2025-10-29

**Soundness:** 2
**Presentation:** 2
**Contribution:** 2
**Rating:** 4
**Confidence:** 3

**Summary:**

This paper addresses three fundamental challenges in federated recommendation (FR): privacy leakage, performance degradation, and high communication overhead. To tackle these issues, we propose FedKDRec, a novel FR framework based on bidirectional knowledge distillation with agent-assisted learning. Unlike conventional federated learning paradigms that exchange model parameters, FedKDRec operates by sharing soft predictions—thereby preserving both data privacy and model parameter confidentiality. To further enhance model performance and communication efficiency, we introduce two key components: (1) a server-oriented that dynamically weights client knowledge in multi-teacher knowledge distillation; and (2) a client-oriented proxy that selectively transmits informative, lightweight samples from the server. Extensive experiments demonstrate that FedKDRec consistently outperforms state-of-the-art knowledge distillation baselines and parameter-transfer FR methods, achieving superior recommendation accuracy with significantly reduced communication costs.

**Strengths:**

The manuscript proposes a novel Federated Recommendation framework that innovatively integrates knowledge distillation, reinforcement learning, and large language model agents into the federated recommendation paradigm, aiming to address three major challenges in this domain. The paper is well-written with a clear structure, making the core ideas accessible and understandable. The comparative experimental section is reasonably designed, encompassing three benchmark datasets and four competitive baseline methods, which provides solid empirical support for the claimed contributions.

**Weaknesses:**

Despite the above advantages of this article, there are still some issues that need attention:
1. The description of state embedding generation in sections 3.2 and 3.3 seems unclear. There is no explanation on how to concatenate these scalars/vectors into a unified dimensional vector, and the paper fails to supplement relevant technical details.
2. Section 3.1 applies a label " swapping" mechanism to protect the original datasets, but there is a logical contradiction. For example, 'high-value predictions' are model outputs, but the real labels are unrelated to user interests. If an item has a high predicted value (such as 0.9), but the true label $r_{i,j}$=0, then it itself is a sample of "prediction error". Exchanging the labels of a 'high values prediction' sample and a sample with a negative label does not generate effective positive samples, but may instead create more noise.
3. The potential performance degradation caused by the "swap" mechanism should be further investigated through ablation studies to validate its effectiveness and necessity.
4. The experiments are solid but somewhat limited in scope. To better demonstrate the generalizability of the proposed method, the authors are encouraged to include results on additional or more diverse datasets (such as ML-10M, Flixster, Douban, etc.) in the final version, if possible.

**Questions:**

1.  Regarding the communication cost:
When comparing communication burden, except for PTF-FedRec, all other methods are traditional benchmark methods (before 2022). It is uncertain whether you have compared the communication load of auxiliary datasets and labels together. If this is not considered, the proposed method will not be convincing in solving the challenge of high communication load. In addition, when the one-hot encode is high-dimensionality, the communication cost may be greater than the model parameters in soft prediction.
In fact, as shown in Table 3, the single-round communication load of earlier studies (such as the benchmark methods studied in 2022) did not have particularly high single round communication loads. In addition, early research on federated learning did face the problem of high communication load, as it required multiple rounds of model parameter sharing and the combination of encryption techniques to prevent gradient leakage attacks. From your research alone, it doesn't seem to involve this challenge?
2.  It would be necessary to add other knowledge distillation methods for comparison.
3.  Will the auxiliary dataset used during training, which mostly consists of item data unrelated to customer interests, have a negative impact on the improvement of model performance?

---

### Meta-Review · Area_Chair_keFE · 2026-01-07

**Summary:**

This paper proposes FedKDRec, a privacy-preserving and communication-efficient federated recommendation (FR) framework. FedKDRec employs a bidirectional knowledge distillation (KD) that allows clients and the server to exchange soft predictions, guided by two reinforcement learning (RL) agents: (1) a server-oriented agent for adaptive multi-teacher client weighting, and (2) a client-oriented actor-critic agent for selective sample transfer. The experiments demonstrate that FedKDRec can outperform state-of-the-art knowledge distillation baselines and parameter-transfer FR methods.

**Reviewer Concerns:**

The concerns are around incremental novelty, weak literature positioning, limited experimental validation, lack of theoretical analysis, and additional costs of two RL agaents, etc. The authors did not perform rebuttal. Some major concerns are summarized as follows:

[Reviewer RyQL] "Claimed novelty is overstated" as "Bidirectional KD in FL is not new", "Multi-teacher KD for FL is well-established", "Mixed-label distillation (hard + soft) is standard practice in KD", "RL-based teacher weighting exists".

[Reviewer 3Cof] The potential mechanism of  label " swapping" is unclear and should be further investigated through ablation studies to validate its effectiveness and necessity.

[Reviewer eQK2] The experiments lack comparisons with more works published in recent venues.

**Reviewer Scores:**

The authors didn't rebuttal.

---

### Decision · Program_Chairs · 2026-01-26

Reject